# Spiking CenterNet: A Distillation-boosted Spiking Neural Network for Object Detection

## Abstract

In the era of AI at the edge, self-driving cars, and climate change, the need for energy-efficient, small, embedded AI is growing. Spiking Neural Networks (SNNs) are a promising approach to address this challenge, with their event-driven information flow and sparse activations. We propose Spiking CenterNet for object detection on event data. It combines an SNN CenterNet adaptation with an efficient M2U-Net-based decoder. Our model significantly outperforms comparable previous work on Prophesee's challenging GEN1 Automotive Detection Dataset while using less than half the energy. Distilling the knowledge of a non-spiking teacher into our SNN further increases performance. To the best of our knowledge, our work is the first approach that takes advantage of knowledge distillation in the field of spiking object detection.

Keywords: SNN, Knowledge Distillation, object detection, event data

## 1 Introduction

In recent years, the integration of object detection capabilities into edge devices has witnessed unprecedented growth, driven by the ever-increasing demand for real-time applications in fields such as automotive and robotics. Edge devices, characterized by their resource-constrained nature, pose unique challenges in terms of computational efficiency and power consumption. Addressing these challenges requires innovative approaches that not only provide accurate object detection but also ensure power-efficiency for operation.

One promising approach for achieving these goals is the utilization of Spiking Neural Networks (SNNs), which are inspired by the communication mechanism of biological neurons. SNNs exhibit inherent power-efficiency as their distinctive feature is event-driven information processing, achieved through all-or-nothing events (spikes) for communication between neurons. This attribute sets SNNs apart from conventional Artificial Neural Networks (ANNs), which primarily rely on non-binary floating-point values (*floats*). Spikes facilitate fast, cost-effective neuron interactions via single-bit electronic impulses, unlike multi-bit data like floats which demand multiple impulses. In addition, sparse binary value transmission conserves energy by keeping most of the neurons inactive during operation.

Similarly to SNNs, event-based cameras also exhibit asynchronous behavior, aligning well with SNNs' processing capabilities. Event-based cameras provide several benefits compared to conventional frame-based cameras: they have an exceptional temporal resolution in microseconds (Gallego et al., 2020), rendering them ideal for applications demanding real-time responsiveness. Furthermore, they excel in energy efficiency, transmitting data only in response to sensory input changes instead of transmitting redundant information as conventional frame-based cameras do. SNNs and event-based cameras work together effectively, providing fast, energy-efficient data processing.

The development of SNN-based object detectors holds substantial promise for advancing the utilization of SNNs in real-time autonomous applications demanding energy-efficient object detection capabilities, unlike the predominant focus of previous SNN research on classification tasks. However, the effective employment of SNNs remains a significant challenge due to the intrinsic difficulty associated with directly training these networks, given their discrete and spiking (i.e., binary) nature and thus non-differentiable activations.

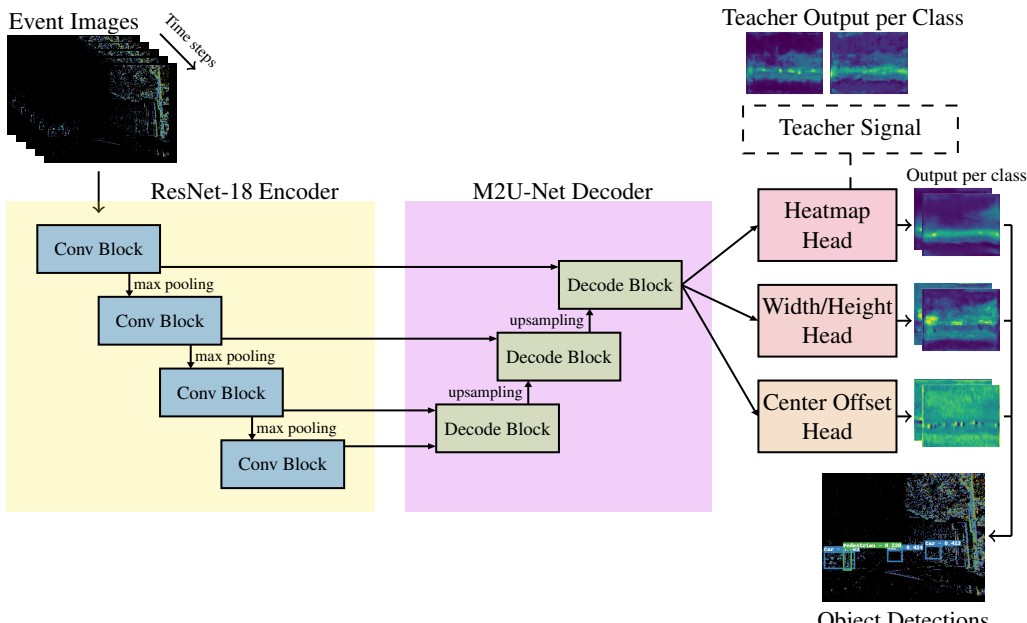

Figure 1: Overview of our spiking object detection model. We combine a ResNet-18 encoder with M2U-Net-based decoding (Laibacher et al., 2019) to feed into CenterNet-based heads (Zhou et al., 2019). We remove any residual connections, and replace all activation functions with Parametric Leaky Integrate-and-Fire (PLIF) neurons. Postprocessing calculates bounding boxes from the head output.

In this work, we propose a novel fully SNN-based object detection framework (see Fig. 1) trained on automotive data recorded by event-based cameras. Our key contributions are as follows:

- We propose a modified, spiking version of the simple and versatile CenterNet architecture (Zhou et al., 2019) which is also - to the best of our knowledge - the first trained SNN detector that does not require costly Non-Maximum Suppression (NMS).

- We replace CenterNet's upsampling by the more efficient modules from M2U-Net (Laibacher et al., 2019) and add binary skip connections between encoder and decoder which improves gradient flow despite the spiking communication.

- To the best of our knowledge, we are the first that utilize Knowledge Distillation (KD) for SNNs in the context of object detection, with the aim of addressing the challenges associated with training efficiency and model generalization.

Our SNN-based object detector outperforms comparable previous work on the challenging GEN1 Automotive Detection (GEN1) dataset by 4 % mean Average Precision (mAP). We show the effectiveness of KD, which improves model performance in terms of mAP by an average of 1.8 % over a baseline SNN. We also show that our model achieves better power efficiency than its non-spiking counterpart and the state-of-the-art SNN-based object detectors.

## 2 RELATED WORK

A major disadvantage of SNNs is training complexity and the spikes' lower information resolution. Firstly, due to the discrete and non-differentiable nature of spikes, back-propagation cannot be performed directly for training. Additionally, the temporal aspect turns SNNs into a type of Recurrent Neural Network (RNN), which is inherently difficult to train (Pascanu et al., 2013). Finally, a series of binary spikes with practical length can only represent a limited amount of different values in contrast to the high precision of floating points.

At the atomic level, there are many different neuron models to use, ranging from the Hodgkin and Huxley model (Hodgkin & Huxley, 1952), over the Izhikevich neuron model (Izhikevich, 2003) to the Leaky Integrate-and-Fire (LIF) model (Delorme et al., 1999). As a good trade-off between complexity and efficiency, we choose the PLIF neuron (Fang et al., 2021), which is a LIF with learnable membrane variables. Due to the aforementioned non-differentiable nature of spikes, special frameworks are needed for training. Some widely-used methods are Spike Layer Error Reassignment in Time (SLAYER) (Shrestha & Orchard, 2018) and surrogate gradient learning (Neftci et al., 2019). While SLAYER uses a temporal credit assignment policy to backpropagate errors to previous layers, surrogate gradients simply approximate the non-differentiable spiking function with a similar differentiable function. We choose surrogate gradients since they enable treating an SNN as a simple RNN, which allows utilization of established learning algorithms such as Backpropagation Through Time (BPTT) (Rumelhart et al., 1986).

There are currently two main directions for implementing spiking object detectors: conversion and training from scratch. Converting the weights of a usually isomorphic non-spiking ANN is popular for creating complex SNNs because it avoids training non-differentiable spiking functions. However, these conversions often result in loss of accuracy, which is why the bulk of work in this direction goes into minimizing conversion loss. SpikingYolo (Kim et al., 2020) is an example of a successful conversion from ANN to SNN for object detection. However, this network requires at least 1 000 time steps to detect objects with acceptable accuracy. Recently, Qu et al. (2023) achieved good accuracy with only four time steps, but they stretch the definition of SNNs. They use non-binary "burst spikes" and spike weighting, which is used to make spike signals more complex, but they disregard the computational impact of it. While conversion enables the reuse of an existing well-trained network, the high number of time steps or more complex spikes both negatively affect the resulting network's efficiency. Furthermore, conversion from a non-recurrent ANN does not allow the resulting SNN to take advantage of temporal event data.

An alternative approach is training SNNs from scratch. Cordone et al. (2022) introduced the first fully spiking SNN for object detection trained on a challenging real-world event dataset. This was an important milestone for SNN research as it showed the feasibility of training from scratch and a low-timestep SNN. Su et al. (2023) developed an even better performing SNN model by introducing a "spiking residual block". However, it includes non-spiking residual connections which violate the SNN's core principle of spiking signals between layers. We also opt to train our SNN object detector from scratch to fully utilize the inherent sparsity of trained SNNs and improve upon the previous fully-spiking standard set by Cordone et al. (2022). Furthermore, we utilize the superior training capabilities of the non-spiking counterpart as a teacher signal for our SNN model through KD.

The idea of Knowledge Distillation is a well-established learning strategy first shown in (Hinton et al., 2015) . It is about improving the performance of a smaller, more efficient "student" network by transferring the knowledge of a larger, more capable "teacher" network as an additional soft learning target to the student network. First examples of using KD for SNNs are limited to simple classification problems. While Tran et al. (2022) use a more traditional KD approach together with SNN-to-ANN conversion, Xu et al. (2023) use a novel approach they call "re-KD" in which they adapt the network structure on-the-fly while distilling knowledge. Our approach, described in Section 3.3, is closer to the former. However, KD in object detection is more complicated than in classification (Chen et al., 2017) and there is - to the best of our knowledge - no previous work of it with SNN detectors.

## 3 METHOD

### 3.1 SPIKING CENTERNET

The main motivation behind constructing our SNN architecture is simplicity, as we find that complex neural network structures, albeit proven for non-spiking ANNs, function worse with spiking activations. For example, highly optimized and complex architectures such as EfficientDet (Tan et al., 2020) suffer from the binarization of feature maps and contain modules such as a singular global feature factor which as a spike may disable a module's output completely. Therefore, we construct our model from two very simple architectures: CenterNet (Zhou et al., 2019) and M2U-Net (Laibacher et al., 2019).

Due to its simplicity and reproducibility, CenterNet (Zhou et al., 2019) has become a very influential object detection model. It features variable backbones and heads for different tasks which encompass 2D and 3D bounding box detection as well as human pose estimation. The key idea of the model is to estimate objects or target points (e.g., joints) as key points (activity blobs centered at target) on 2D classification heatmaps (one for each class, cf. Fig. 4). These heatmaps divide the input image into a grid of variable size. To balance the grid's coarseness, an offset regression with similar shape is also produced. Additionally, depending on the task, bounding box width/height regression may also be used. These predictions are made by individual heads which take feature maps of roughly the same size as the input. This makes the backbone structure similar to a segmentation network with an encoder and a decoder part.

The final bounding box predictions are produced by an extraction of local maxima from the heatmap, which replaces the typical NMS (Bodla et al., 2017) found in other detection models. This allows us to fully utilize the time dimension and produce several outputs with spiking CenterNet heads with a hidden spiking layer of 64 channels for the heatmap, offset regression and bounding box width/height regression (Zhou et al., 2019), rather than only aggregating features over time and performing a single-step detection as done by Cordone et al. (2022). In our work, we take the mean of each head's output over all five time steps to produce a final, more robust output. This allows the model to be independent of the specific number of time steps and generate results with fewer time steps if needed (see Fig. 3).

Among the different backbones, we opt for ResNet-18 (He et al., 2016) due to its simplicity and relatively small size. We adopt the SpikingJelly's implementation (Fang et al., 2020) of the network and replace the classic ReLU activation with SpikingJelly's PLIF neuron throughout the network, including the decoder. We replace the first convolutional layer to adapt to the number of input channels depending on the data (i.e., 4 channels for event data, see Section 4.1.1).

### 3.2 M2U-Net Decoding

M2U-Net (Laibacher et al., 2019) is a popular small segmentation network with 0.55M parameters. It features an encoder-decoder structure similar to the ResNet-18 based backbone in CenterNet (Zhou et al., 2019). However, M2U-Net uses a static upsampling step rather than weight-based deconvolution as in CenterNet's decoding.

Originally, CenterNet's ResNet-18-based version uses so-called *transposed convolutions* or *deconvolutions* (Zeiler et al., 2010) to increase the feature maps' size before feeding them to CenterNet's heads. However, this relies heavily on the ability of the network to compress spatial information in the rather low-resolution, but high-dimensional feature maps of the encoding steps. It also requires the network to learn meaningful deconvolutional weights. Since SNNs are quite limited in feature map output due to their binary nature and generally work better with fewer tunable weights, we instead choose a decoding strategy based on M2U-Net.

M2U-Net's decoding (Laibacher et al., 2019) is particularly suited for SNNs. Its skip connections between encoder and decoder allow the SNN to retain important high-level information more easily, and its simple weightless upsampling spares the SNN from unnecessary weights. To do so, we add M2U-Net's upsampling blocks and connect the encoder blocks' outputs with the decoder blocks of the same input size (see Fig. 1). We replace the ReLU activations with PLIF neurons. However, in M2U-Net's *Inverted Residual Block*, we drop the activation function between the depth-wise and the point-wise linear convolutions since we can merge them for inference, thus eliminating non-spike signals between these layers. Furthermore, we remove the identity connection in the Inverted Residual Connections, as the summation of the identity and residual and the resulting non-binary values violates the idea of a (binary) SNN. We find that a Boolean OR-operation as a binary alternative does not improve the result, and instead, we decide to drop the identity connection entirely.

In this way, we create a combination of two networks that we call Spiking CenterNet, which offers a flexible arrangement consisting of simple, SNN-compatible building blocks.

### 3.3 Knowledge Distillation for SNNs

As the isomorphic, non-spiking counterpart of our SNN model performs better than the SNN, we try to distill knowledge from this non-spiking version to the spiking one. Our approach is straightfor-

Table 1: Results on the GEN1 dataset (De Tournemire et al., 2020).

| Model | #Params (Millions) | mAP best | mean | T | Energy (mJ) |
|---|---|---|---|---|---|
| Non-spiking ANNs: | | | | | |
|    HMNet-L3 (Hamaguchi et al., 2023) | - | 0.471 | - | - | - |
|    Ours (ANN teacher) | 12.97 | 0.278 | 0.275 | 1 | 28.214 |
| SNNs: | | | | | |
|    DenseNet121-24+SSD (Cordone et al., 2022) | 8.2 | 0.189 | - | 5 | 10.485 |
|    EMS-Res10-YOLO (Su et al., 2023) [1] | 6.2 | 0.267 | - | 5 | - |
|    EMS-Res18-YOLO (Su et al., 2023) [1] | 9.3 | 0.286 | - | 5 | 1.965 [2] |
|    Ours (no KD) | 12.97 | 0.223 | 0.205 | 5 | 3.096 |
|    Ours (with KD) | 12.97 | 0.229 | 0.223 | 5 | 4.995 |

1 Su et al. (2023) use non-spiking residual connections.

2 Excludes energy consumption from the first coding layer.

ward: First, the non-spiking teacher is trained separately and then the weights are frozen during the training of the SNN. For each time step during the latter, we pass the same input to both the SNN and the teacher. Finally, the teacher's output is used as a soft target signal to calculate the mean squared error:

$$L_{\text{teach}} = \frac{1}{T} \sum_{t=0}^{T-1} \sum_{p \in \text{Pixels}} \{o_p(t) - \hat{o}_p(t)\}^2, \tag{1}$$

where $o_p(t)$ is the output of the $p$-th pixel of the SNN model at time $t$ and $\hat{o}_p(t)$ is the corresponding teacher output. With this we arrive at an overall loss of:

$$L = L_{CN} + \alpha \cdot L_{\text{teach}}, \tag{2}$$

where $L_{CN}$ is the CenterNet loss as in Zhou et al. (2019). We choose $\alpha = 1$ after initial experiments.

## 3.4 MEASURING ENERGY CONSUMPTION OF ANNS AND SNNS

One of the most important benefits of SNNs—compared to ANNs—is their energy efficiency. Measuring this advantage, however, is nontrivial if SNN hardware is not yet available. For an ANN, the number of synaptic operations per layer can be calculated from the architecture of the convolutional and linear layers, where a Multiply-Accumulate Computation (MAC) takes place per synaptic operation, multiplying each non-spiking activation with the respective weight before adding it to the internal sum. For an SNN executed on a neuromorphic processor, an Accumulated Computation (AC) is performed per synaptic operation only upon the receipt of incoming spikes (Chen et al., 2023) where the corresponding weights only need to be accumulated at the target neuron. Therefore, the total number of AC operations is calculated by a product of the average firing rate for a particular layer and the corresponding number of synaptic operations. However, many modern SNNs trade efficient energy consumption for more accuracy by also using non-spike operations that also results in MACs. According to Chen et al. (2023), the theoretical computational consumption can be determined by the number of AC and MAC operations:

$$E_{\text{SNN}} = T \cdot (f \cdot E_{\text{AC}} \cdot O_{\text{AC}} + E_{\text{MAC}} \cdot O_{\text{MAC}}). \tag{3}$$

Here, T is the simulation time and $f$ is the average firing rate. $E_{\text{AC}}, E_{\text{MAC}}$ and $O_{\text{AC}}, O_{\text{MAC}}$ are the energy consumption and number of operations for AC and MAC, respectively. We assume energy consumption values of $E_{\text{AC}} = 0.9\text{pJ}$ and $E_{\text{MAC}} = 4.6\text{pJ}$ based on current 45 nm technology following related works (Horowitz (2014), Su et al. (2023)).

## 4 EXPERIMENTS

### 4.1 IMPLEMENTATION DETAILS

#### 4.1.1 DATA

We train and evaluate all our models on the *GEN1 dataset* (De Tournemire et al., 2020). It consists of 39 hours of recordings with the $304 \times 240$ pixel GEN1 sensor. Its event-based nature makes it particularly useful for training and evaluating SNNs as it natively provides spike-like and sparse input. GEN1 features an impressive number of 255,000 annotations for the two classes cars and pedestrians. These qualities made it an established benchmark for SNN-based object detectors (Cordone et al., 2022; Su et al., 2023). Fig. 2 shows selected scenes.

Following the procedure of Cordone et al. (2022), we sample $100 \, \text{ms}$ of events preceding every annotation and split it into binary voxel cubes of 5 time steps, with each split into two micro time bins that are processed simultaneously. Together with the polarity of events, this gives us $2 \times 2 = 4$ input channels. However, as our non-spiking teacher model only uses one time step, we instead sample $20 \, \text{ms}$ for its training to keep the information per time step similar.

#### 4.1.2 HYPERPARAMETERS

We train both our spiking and non-spiking models with the AdamW optimizer with a weight decay of 1e-4. While the non-spiking model uses a learning rate of 1e-3, the SNN models use 1e-4. All models use cosine annealing learning rate scheduler that reduces the learning rate to 1e-5. We clip our gradients at 1 to avoid exploding gradients. Due to faster convergence, the non-spiking model is trained for just 20 epochs while the spiking models are trained for 50 epochs. We initialized all but the output convolutions with the Kaiming Uniform method and zero bias. The heatmap head's last convolution's bias was initialized as -2.19 following Zhou et al. (2019) as it results in 1.0 after softmax activation. The size regression and offset heads' biases were initialized as 0.15 and 0.5 based on empiric convergence after some initial experiments. The corresponding convolution heads were initialized with normal distribution with standard deviation 1e-3.

#### 4.1.3 TESTING

Our main performance metric is the COCO mAP (Lin et al., 2014) calculated over 10 IoU values ([.50:.05:.95]) as it is the de-facto standard metric for object detection. Unlike previous works (Cordone et al., 2022), we focus on the *mean* performance of five trained model instances with different seeds as a more robust measurement and report the maximum, i.e., best performing model only for comparison.

Additionally, we aim to quantify the computational performance of our models as this is the main motivation behind the development of SNNs. In order to do so, we report the following metrics:

- *Number of parameters:* As in all neural networks, the number of parameters correlates with energy consumption. Additionally, embedded (neuromorphic) hardware as the desired deployment environment often features limitations on network size.

- *AC & MAC:* We report both the AC and MAC operations as measured by the SyOPs python library (Chen et al., 2023) to calculate the theoretical energy needs for both the non-spiking and spiking models.

- *Firing rate:* We also record the firing rate of the SNNs by calculating the proportion of active neurons (i.e., spikes) among all neurons (i.e., possible spikes) averaged over the test set and time steps (cf. *sparsity* in Cordone et al. (2022)).

To calculate the energy consumption of the SNNs, we mainly use the open-source tool *syops-counter* provided by Chen et al. (2023). As depthwise-separable convolutions and Batch Normalization (BN) layers after convolutions are useful for training, but introduce float values and thus MAC operations, we merge them according to Rueckauer et al. (2017) before counting energy. We compute the energy consumption based on the validation split of the GEN1 dataset. The results are reported in Table 1 (energy) and Table 2 (#operations, firing rate).

Table 2: Total number of operations over time and firing rate.

| Model | MACs | ACs | Firing rate |
|---|---|---|---|
| DenseNet121-24+SSD (Cordone et al., 2022) | 0 | 11.65G | 37.20 % |
| EMS-Res18-YOLO (Su et al., 2023) | - | - | 20.09 %[1] |
| Ours (ANN teacher) | 6.13G | 0.018G [2] | 100.0 % |
| Ours (no KD) | 0 | 3.44G | 10.8 % |
| Ours (with KD) | 0 | 5.55G | 17.4 % |

1 Excludes energy consumption from the first coding layer.

2 Stem from binary event data input.

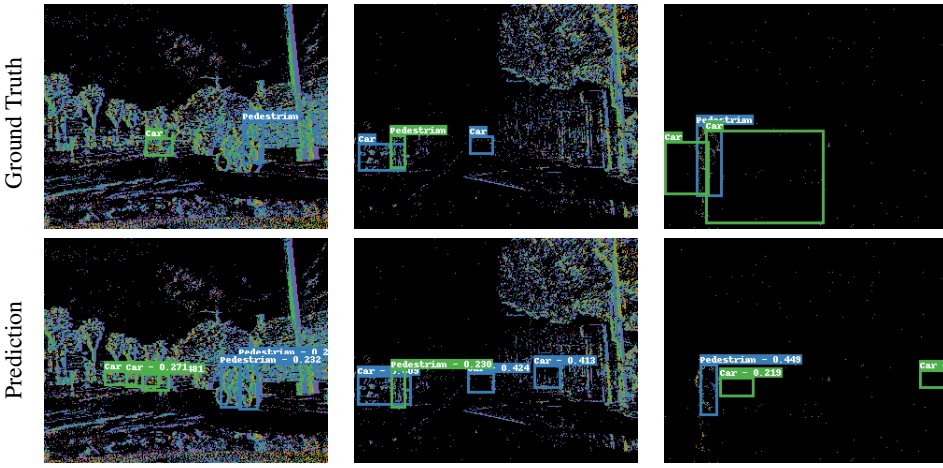

Figure 2: Prediction of our best SNN model (bottom) and ground truth (top) for selected scenes of the GEN1 dataset. The different pixel colors indicate the two micro time bins with each two polarities of brightness change, resulting in four input channels (cf. Section 4.1.1). Note that targets might be invisible if there is no camera or object motion.

## 4.2 RESULTS

We report in Table 1 results for three models: Our non-spiking ANN baseline (and teacher), the isomorphic SNN model without KD, and the SNN model trained with KD . Our results show that our simplified SNN model with KD reaches a competitive mAP of 0.229 (maximum) and 0.223 (mean), outperforming previous comparable models (Cordone et al., 2022) by a significant margin of 4 % mAP. Fig. 2 shows object detections for selected scenes.

The results in Tab.1 also suggest that our KD approach makes the SNN model consistently perform better and less reliant on outliers with a mean mAP difference of +1.8 %. Furthermore, we discover, that despite the higher number of parameters in our model, it is sparser and more energy efficient than comparable models (cf. Table 1 & Table 2). However, we observe that the recent work of Su et al. (2023), who mix non-spiking structures into their SNN model, still performs better in terms of mAP performance.

Regarding energy consumption, both our baseline SNN and KD-based model significantly outperform the 10.485 mJ of the model in Cordone et al. (2022) with 3.096 mJ and 4.995 mJ, respectively, for the entire sequence of 5 time steps. Su et al. (2023) report a lower energy consumption, but ignore the initial convolutional layer's computational impact, making comparison to our results difficult.

## 4.3 ABLATION STUDIES

In order to explore the capabilities of our SNN models on working with fewer time steps, we first select the best instance of our models with and without KD based on the results on the validation subset of the GEN1 dataset. We then modify two parameters: The sequence length, i.e., the number

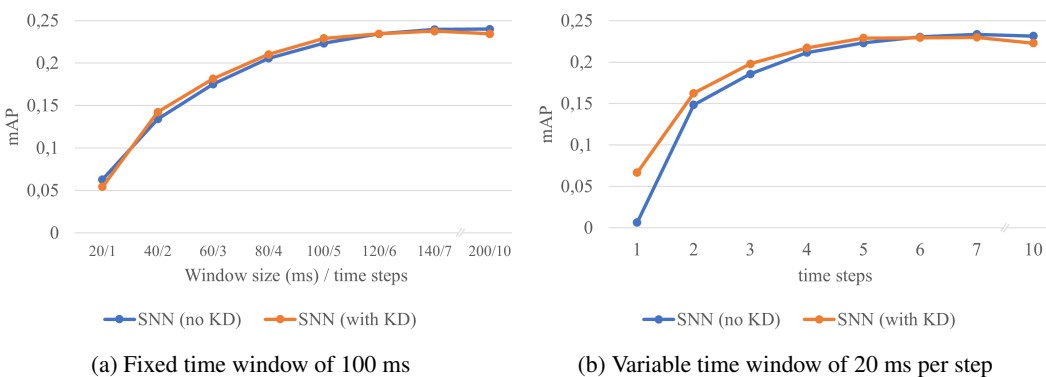

Figure 3: Impact of the number of time steps in evaluation with a fixed (a) and variable (b) time window for sampling events. Shown is mAP of our best SNN models on the GEN1 dataset (De Tournemire et al., 2020).

of time intervals the input is divided into, and the sample window size, i.e., the time window in milliseconds we sample before each ground truth bounding box (see Section 4.1.1). We evaluate each 5-time-steps-trained model with a different number of time steps, ranging from 1 to 10. We do so twice: with a fixed time window of 100 ms, i.e. the same information is compressed into fewer time steps of longer duration, and with a decreasing time window in which each time step has a fixed duration of 20 ms. The results are shown in Fig. 3. We find that albeit performance unsurprisingly drops with the number of time steps, performance with 4 time steps still beats previous models with 5 time steps (Cordone et al., 2022) and even 3 time steps still deliver decent performance. While simply dividing the same time window into more than 5 time steps does not improve performance significantly, a bigger time window does help slightly. However, past 140 ms a bigger time window does not help either.

## 5 DISCUSSION

The main motivation behind our work is introducing a simple, well-performing architecture which strictly adheres to the definition of an SNN. Some previous works try to define complex, weighted signals as spikes while ignoring the additional computational cost these non-binary "spikes" introduce (Qu et al., 2023). Other works such as Su et al. (2023) hide additional non-spiking operations within "spiking" blocks: Within their *EMS-ResNet*, non-binary values are added, max-pooled and transmitted as residual connections over long distances, rather than cheap binary spikes. These not only incur additional costly MAC operations, it is also unclear whether spiking neuromorphic hardware will be able to support such complex neuron blocks.

In light of these concurrent works, it is our desire to keep our architecture as simple as possible and limit non-binary values to just the interface between convolutional and spiking activation layers as well as the final output, where the lack of subsequent neurons make spikes less valuable. Our M2U-Net-based (Laibacher et al., 2019) skip connections, connecting the encoder and decoder part of our backbone, merely transport sparse binary spikes. These are then concatenated and thus do not add MAC operations. We found that these skip connections are quite vital for gradient flow and enable the deep structure of the network; prior to adding them, the model would not learn at all on the GEN1 dataset. Furthermore, our Spiking CenterNet is the first SNN detector without the expensive NMS, which also allows us to fully utilize the time dimension of the output. Lastly, Spiking CenterNet, due to its task-specific heads, is like the original CenterNet also easily expandable to other difficult tasks like 3D bounding box detection and pose estimation, which are not yet explored with SNNs.

We are also—to the best of our knowledge—the first to utilize Knowledge Distillation for training a spiking object detector. Since we observe that the best non-spiking ANNs still outperform SNNs by a wide margin (see Table 1), it was our hope to transfer this performance to our SNN model. We observe that KD indeed improves the result, especially mean performance. It can thus be used to make the training more consistent. We also observe that a KD-boosted SNN seems to produce

smoother, and according to mAP better, heatmap outputs (see Fig. 4). However, it also increases the number of spikes, therefore presenting a trade-off between performance and energy consumption.

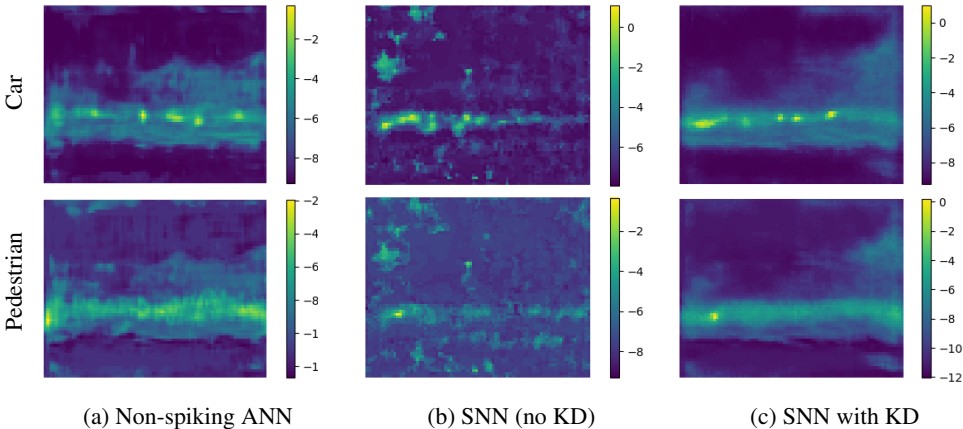

| (a) Non-spiking ANN | (b) SNN (no KD) | (c) SNN with KD |

Figure 4: Output of heatmap head (see Fig.1) averaged over time steps of the three evaluated models. Knowledge Distillation from the non-spiking ANN teacher to the SNN results in a less sparse, but smoother and ultimately better heatmap.

Our evaluation reveals that despite our higher number of parameters, our SNN models actually use fewer spikes than comparable models (Cordone et al. (2022), see Table 2). This results in a lower proportion of active neurons and thus lower firing rate. However, although a low firing rate is generally considered good (Cordone et al., 2022), it might also indicate an unnecessarily large network. Especially in light of possible limitations regarding neuron numbers in neuromorphic hardware, eliminating neurons which are inactive most or all of the time is logical. Nevertheless, first solving the object detection task at all to a satisfiable degree, which is a challenge in and of itself, is the prime priority of our work.

Finally, our findings in evaluating our SNN models with fewer time steps indicate that our model can produce good results within a shorter time window than it has been trained for. This seems to confirm our decision of taking the mean of the overall network output over time as it makes the model less reliant on producing the correct output at all five time steps. Of course, during real-time inference there are no time steps, but our hope is that in this case the neurons' leaky membrane constant would smooth over the asynchronous events.

## 6    CONCLUSION AND FUTURE WORKS

We presented a new, versatile SNN architecture for object detection in the form of our *Spiking CenterNet*, consisting solely of simple building blocks and not requiring expensive NMS. Moreover, we are the first (to the best of our knowledge) to employ Knowledge Distillation for spiking object detection, which improves our baseline SNN model significantly. We observed that our SNN not only beats comparative previous work by 4 mAP points, but also uses less than half the energy. We demonstrate in our work that it is possible to push the performance of SNNs without stretching the definition of what constitutes an SNN. Furthermore, we show that even the simplest form of KD can work for spiking object detection. More sophisticated KD versions (e.g., for intermediate features) or more complex ANN teachers could be investigated in future works. We also plan to extend our approach to both RGB data input and also to different tasks such as 3D bounding box and human pose estimation, comparable to the original CenterNet (Zhou et al., 2019).

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
