# OpenReview forum: "Spiking CenterNet: A Distillation-boosted Spiking Neural Network for Object Detection"
_ICLR.cc/2024/Conference — Submitted to ICLR 2024_

### Official Review · Reviewer_WJNq · 2023-10-24

**Soundness:** 3 good
**Presentation:** 3 good
**Contribution:** 2 fair
**Rating:** 6
**Confidence:** 4

**Summary:**

The paper presents Spiking CenterNet, an SNN for object detection. It employs knowledge distillation from a non-spiking SNN to improve the accuracy. The implementation of the proposed method for the Prophesee’s GEN1 Automotive Detection Dataset shows better results than related works.

**Strengths:**

1. The tackled problem is relevant to the community.

2. The technical descriptions are clear and comprehensive.

3. The results show better results than prior SNNs.

**Weaknesses:**

Some aspects need to be clarified. Please refer to the questions below.

**Questions:**

1. Please highlight more clearly the differences between Cordone et al. (2022) and the proposed method (other than applying knowledge distillation).

2. Please discuss what are the challenges of applying knowledge distillation for SNN object detection, compared to existing knowledge distillation methods between other types of networks.

3. Please provide more details (setup and tool flow) for the implementation of the proposed method on the neuromorphic hardware.

4. If possible, please provide the source code for reviewers’ inspection during the rebuttal.

---

> ### Author Response · Authors · 2023-11-17
>
> Thank you for your review. Our main difference to the work of Cordone et al. besides the use of KD is our CenterNet-based, anchor-less object detection via a heatmap which is particularly suited to spikes (and also enables an easy application of KD). It is also composed of very simple building blocks (see more below). We will try to highlight this stronger in our work.
>
> Thank you for pointing out that the challenges of KD in SNN-based object detection are not clear enough in our work; there are a variety of them, and we will try to showcase them better. We should point out, however, that all previous works of KD with SNNs focused on classification as the topic of SNN-based object detection even without KD is not yet very well researched.
>
> We would like to integrate our model into suitable spiking neuromorphic hardware. That is our main motivation behind choosing relatively simple building blocks of convolution layers immediately followed by spiking neurons to ensure we do not propagate float values over long distances such as the works of Su et al. (2023). However, the only commercially available SNN chip, the Akida Brainchip, is not only aimed at a different neural network framework, but also focuses on running converted, rather than trained SNNs and does not offer many details about the underlying spiking dynamics. Nevertheless, we are in close contact with an institution that is developing their own SNN chip and future works of ours aim at running our models on their hardware."

---

> > ### Comment · Reviewer_WJNq · 2023-11-23
> > **Response to Authors' Rebuttal**
> >
> > Thank you for your responses. Considering together the other reviewers' comments and responses, my score is confirmed.

---

### Official Review · Reviewer_DzYu · 2023-10-27

**Soundness:** 3 good
**Presentation:** 2 fair
**Contribution:** 2 fair
**Rating:** 5
**Confidence:** 4

**Summary:**

This paper proposes Spiking CenterNet for object detection on event streams. It combines an SNN CenterNet adaptation with an efficient M2U-Net-based decoder. This work is the first approach that takes advantage of knowledge distillation for object detection using SNNs.

**Strengths:**

1) The topic of distillation-boosted SNN for object detection is very interesting and attractive.

2) This paper replaces CenterNet’s upsampling by the more efficient modules from M2U-Net (Laibacher et al., 2019) and add binary skip connections between encoder and decoder, which improves gradient flow despite the spiking communication.

3) This work utilizes Knowledge Distillation (KD) for SNNs in the context of object detection.

**Weaknesses:**

1) The innovative KD in this paper has little effect on the accuracy, and only improves by 0.006. Related knowledge is not explained clearly, such as event-based object detection methods. The advantages of using CenterNet, such as the need for an NMS, are not explained.

2) The authors should explore deeper backbone for the proposed framework, such as ResNet50, ResNet101. Whether the effect is better than shallow network?

3) The details of the proposed method are not clear. For example, how is the identity mapping between encoder and decoder implemented, and how is it different from the implementation in ANN?

4) The relevant work is not fully introduced, and KD has little effect on performance improvement, which brings large energy consumption.

**Questions:**

See Weaknesses

---

> ### Author Response · Authors · 2023-11-17
>
> Thank you for your review. We have seen that our Knowledge Distillation(KD) approach does indeed improve performance significantly by a mean difference of 1.8 mAP. A single outlier of our 5  training runs without KD with a performance close to the KD-based trainings may mask that effect, but highlights why the maximum value is not very reliable. The only reason we include it is to keep some comparability to previous works which only report the maximum performance. We do acknowledge that we cut back on explaining some other advantages of CenterNet, which are not related to its suitability for SNNs, due to the page limitation. We will try to remedy this.
>
> Exploring the effects on deeper networks is of interest to us but they introduce significant challenges. First, due to the additional time dimension the hardware requirements rise significantly, making training iterations more difficult. Second, the error due to the approximate nature of surrogate gradients occurs in each layer and thus accumulates with increased depth of the network. Additionally, the spike-based nature makes sustaining gradients over deep networks difficult. This means deeper SNNs are generally harder to train by sheer computational limits, even without factoring in the inference of a teacher model. However, it is true that KD might be help with some of these issues.
>
> Our chosen identity mapping between encoder and decoder is the same for both the ANN and SNN as the skip signal is concatenated to the decoder signal. It thus does not change the binary nature of the SNN skip signal. In contrast, residual identity connections are usually added to a module's output to let it learn the residuals, and thus introduces non-binary results after that addition which violate the binary nature of our spikes. Boolean alternatives which would preserve the binary nature severely limit the advantage of these connections and were not helpful in our initial trials.
>
> We observed that while our KD did increase the energy consumption, the resulting network is still vastly more energy-efficient than the ANN alternative based on the number of MAC and AC operations.

---

> > ### Comment · Reviewer_DzYu · 2023-11-21
> > **I stuck with the initial score.**
> >
> > The author's responses address my concerns, but there's room for optimization in the network structure's design and writing.

---

> > > ### Author Response · Authors · 2023-11-21
> > > **New paper version online**
> > >
> > > Thank you for taking time to engage our comments. We have uploaded a new version of our paper in which we tried to engage the reviewers concerns better. We hope that may change your mind, but regardless of the outcome, we really appreciate the constructive feedback.

---

### Official Review · Reviewer_Dq9d · 2023-10-29

**Soundness:** 2 fair
**Presentation:** 3 good
**Contribution:** 2 fair
**Rating:** 3
**Confidence:** 4

**Summary:**

The paper introduces "Spiking CenterNet," a novel object detection approach that leverages the energy-efficiency of Spiking Neural Networks (SNNs). Positioned as a solution for the growing demand in edge devices and self-driving cars, this method combines an SNN-based adaptation of CenterNet with an M2U-Net-based decoder. Significantly, the authors incorporate Knowledge Distillation to further enhance the performance of their SNN, making their model stand out in object detection tasks using event data. The primary contribution lies in merging the benefits of SNNs with knowledge distillation to optimize object detection in energy-constrained environments.

**Strengths:**

The integration of an SNN adaptation of CenterNet with an efficient M2U-Net-based decoder is a novel approach in the object detection domain.
The model not only addresses the energy efficiency challenge but also outperforms comparable object detection models on event data.

**Weaknesses:**

1.	In principle, there’s no new architecture built for object detection
2.	The preprocessing of event-data makes it quite similar to binarized conventional videos, instead of digging into the intrinsic benefits of asynchronous properties.
3.	The improvements made by Knowledge Distillation are quite limited, while the whole paper emphases on KD a lot. More importantly, since the idea of KD is guiding SNN models by ANN models, how to obtain the unique advantages of SNNs from KD.
4.     The sparsity listed in table 2 is not the sparsity but the density (higher sparsity means sparser).

**Questions:**

See the weaknesses, also:

1. what's the reason for choosing the architecture? Is that possible to employ other kinds of structures?
2.  How to guide an efficient SNN by an ANN teacher, in terms of the spatio-temporal representation ability, the non-linear firing dynamics etc?
3. Why emphases on KD a lot, given the benefits of KD are quite limited.

---

> ### Author Response · Authors · 2023-11-17
>
> Thank you for your review. We are proposing a new architecture as we not only created a spiking version of CenterNet, but also fundamentally changed its decoder part with structures inspired by M2U-Net. We focused on keeping the basic structures simple to maximize the likelihood that it will be supported by future neuromorphic SNN hardware.
>
> We acknowledge that the preprocessing we took from Cordone et al. 2022 makes less use of the asynchronous properties of the event data. However, our main goal was solving this very difficult task (that only one previous work with a pure SNN approach has addressed) and improving on the previous results at all while developing a more flexible architecture.
>
> SNN training due to the on-off behaviour of the spiking neurons is notoriously unstable and difficult. We have seen that while a single training run without Knowledge Distillation (KD) came close to our KD-based training, in the vast majority of cases our KD-based training was significantly better. This can be seen in the more reliable mean performance over our 5 trainings runs, rather than the maximum prone to outliers. We also see that while KD with an ANN teacher decreases the energy efficiency of the SNN, the SNN is still vastly more efficient than the non-spiking ANN as the sparse nature of the SNN has not changed much.
>
> Regarding the 'sparsity', we have used the interpretation of this term established in previous work (Cordone et al, 2022) to avoid confusion when comparing the works. We acknowledge, however, that we find this interpretation also rather unintuitive and will change it accordingly.

---

### Official Review · Reviewer_gBpC · 2023-10-31

**Soundness:** 1 poor
**Presentation:** 1 poor
**Contribution:** 1 poor
**Rating:** 1
**Confidence:** 5

**Summary:**

The paper does knowledge distillation on SNNs

**Strengths:**

Not applicable

**Weaknesses:**

This paper has no contributions besides just applying knowledge distillation on SNN. Further there are many works that have shown ANN-to-SNN distillation can help. I recommend the authors to look at all the SNN work out there that focus on improving SNN performance using interesting SNN optimization techniques such as those from the research group of Priya Panda, Guoq Li, and many others....

**Questions:**

See weaknesses above

---

> ### Author Response · Authors · 2023-11-17
>
> Thank you for your review. We would like to point out three things: First, we introduce a novel SNN architecture based on several suitable ANN models which significantly outperforms comparable purely SNN-based approaches. Second, we also are, to the best of our knowledge, the first ones to utilize Knowledge Distillation for SNN-based object detection, a vastly more complex problem than the image classification task that is indeed well-researched with both SNNs and Knowledge Distillation. Lastly, we are actually citing three of Guoqi Li's works. Among other things, we use the SpikingJelly framework developed by his group as the core of our SNN models (see Section 3.1). Please let us know whether this clears up your concerns.

---

### Author Response · Authors · 2023-11-17

Please excuse our late replies. We wanted to give one comprehensive reply to each reviewer that addresses all points simultaneously, rather than a multitude of iterative comments, to lessen the workload for every reviewer. Individual replies to reviewers can be found below. We are very thankful for the in-depth analysis and helpful suggestions.

---

### Meta-Review · Area_Chair_8AQp · 2023-12-07

**Metareview:**

The paper proposes Spiking CenterNet for object detection on event data by combining an SNN CenterNet adaptation with an efficient M2U-Net-based decoder. Knowledge distillation has been leveraged to improve  training efficiency and model generalization.

Reviewers are mostly concerned about the lack of novel technical contributions. They have pointed out some relevant existing works that the paper should discuss and compare with in detail. Furthermore, the improvement introduced by the knowledge distillation appears to be rather limited, making the effectiveness of this key component much less convincing.

**Justification For Why Not Higher Score:**

Knowledge distillation as a key component of the proposed Spiking CenterNet only brings marginal performance improvement and the overall technical contribution appears to be limited.

**Justification For Why Not Lower Score:**

N/A

---

### Decision · Program_Chairs · 2024-01-16

Reject